# A Review on Machine Learning, Artificial Intelligence, and Smart Technology in Water Treatment and Monitoring

Matthew Lowe [1], Ruwen Qin [1] and Xinwei Mao [1,2,*]

1 Department of Civil Engineering, Stony Brook University, Stony Brook, NY 11794, USA; matthew.lowe@stonybrook.edu (M.L.); ruwen.qin@stonybrook.edu (R.Q.)
2 New York State Center for Clean Water Technology, Stony Brook University, Stony Brook, NY 11794, USA
* Correspondence: xinwei.mao@stonybrook.edu

**Abstract:** Artificial-intelligence methods and machine-learning models have demonstrated their ability to optimize, model, and automate critical water- and wastewater-treatment applications, natural-systems monitoring and management, and water-based agriculture such as hydroponics and aquaponics. In addition to providing computer-assisted aid to complex issues surrounding water chemistry and physical/biological processes, artificial intelligence and machine-learning (AI/ML) applications are anticipated to further optimize water-based applications and decrease capital expenses. This review offers a cross-section of peer reviewed, critical water-based applications that have been coupled with AI or ML, including chlorination, adsorption, membrane filtration, water-quality-index monitoring, water-quality-parameter modeling, river-level monitoring, and aquaponics/hydronics automation/monitoring. Although success in control, optimization, and modeling has been achieved with the AI methods, ML models, and smart technologies (including the Internet of Things (IoT), sensors, and systems based on these technologies) that are reviewed herein, key challenges and limitations were common and pervasive throughout. Poor data management, low explainability, poor model reproducibility and standardization, as well as a lack of academic transparency are all important hurdles to overcome in order to successfully implement these intelligent applications. Recommendations to aid explainability, data management, reproducibility, and model causality are offered in order to overcome these hurdles and continue the successful implementation of these powerful tools.

**Keywords:** artificial intelligence; water treatment; machine learning; hydroponics; Internet of Things; monitoring





## 1. Introduction

The need for sustainable and clean water access is important to water- and wastewater-treatment plants and many other natural and industrial systems that rely on the resource. In addition to meeting the needs of consumers and providing necessary quality-of-life upgrades to infrastructure, treatment plants must also contend with complex regulatory measures to meet increasing standards of quality [1]. This is only confounded as countries continue to experience heavily polluted waterways, affecting human life as well as aquatic and terrestrial life. As countries continue to industrialize and modernize, these issues are observably worsening [2]. Researchers globally have studied methods to optimize, remediate and improve our applications involving water usage [3–5]. For many, there has been sufficient attention toward creating and simulating optimized, cost-effective, and intelligent models to aid in this challenge.

Artificial intelligence (AI) refers to the idea of endowing algorithms with the ability to perform tasks and make inferences that would require an intelligent human in the same position, while machine learning (ML) relates to intelligent systems that can adapt their behavior during the system-training stage to newly provided information [6]. ML, being

a bottom-up mathematical model relying on historical datasets, excels in its correlative abilities and is able to make inferences in systems with complex mathematical origins that even humans have difficulty understanding and applying [7]. Their ability to aid and support understanding is majorly responsible for the rise in AI and ML applications in academic communities of all subjects and industries [8], including areas studying water treatments such as coagulation [9] and chlorination dosing, membrane-filtration modeling, adsorption processes, and natural-systems monitoring such as river-quality modeling, and agricultural system health.

Emerging AI and ML, coupled with smart technology, are filling a niche in water applications that were previously underserved by traditional techniques and thinking. Some reports estimate that AI expenditures in the water industry will account for almost 10% of the investment of over $90 billion that is expected to mature by 2030 [10]. In water applications, AI, ML, and smart technologies are expected to model and overcome complex and difficult issues through their generalization, resilience, and relative ease of design to achieve cost savings and optimize processes [11,12]. Water applications that have seen notable ML utilization include water and wastewater treatment, natural-systems monitoring, and precision/water-based agriculture. These industry studies have been observed to rely on numerous ML techniques, with the most commonly used including artificial neural networks (ANNs), recurrent neural networks (RNNs), random forest (RF), support vector machine (SVM), and adaptive-neuro fuzzy inference systems (ANFISs) with occasional AI methods including fuzzy inference systems (FISs). There have also been some applications involving hybrid techniques that marry two ML systems, including ANN–RF and SVM–RF. Studies have recorded success in their applications of both AI and ML in water-based usages for optimizing modeling processes.

While these successes have been noted, AI and ML applications are not without their challenges that must be overcome before widespread implementation occurs. This review offers a cross-section of mostly ML techniques, with some AI and smart technologies, that have been applied in water-based applications to optimize and model water- and wastewater-treatment processes (including chlorination, adsorption, and membrane-filtration processes), natural-systems monitoring, including dissolved-oxygen monitoring, water-quality-index monitoring and water-level monitoring, and water-based agriculture including hydroponics and aquaponics. This review is not intended to be all-encompassing of AI, ML, and smart-technology applications in water-based studies but rather to show the current pulse of many of these important published works. While all the categories include descriptions of the model(s) used, as well as input and output parameters, the individual analyses also include topics of specific relevance. As the reviewed journals' successes are noted, so too are the shortcomings that were nearly ubiquitously discovered. Challenges and their implications to future success, along with recommendations and conclusions, are included to map these common shortcomings, and to offer a path beyond them.

## 2. Review Search Criteria and Methodology

This review included a search of peer-reviewed publications on Web of Science to evaluate the application of artificial intelligence, machine learning, and smart technology in water treatment and monitoring. Searches were separated into four broader categories: "water treatment", "water systems monitoring", "hydroponics" and "aquaponics". Hydroponics and aquaponics are included as separate search results in order to identify water-specific applications of AI, ML, and smart technologies that rely on nutrient/environmental control and monitoring. These broader categories were matched with specific keywords to better define the search results including "artificial intelligence", "machine learning", and "smart technology". The search was limited to peer-reviewed publications that included specific mentions of methods and models used. Other review publications were not included in this paper. The literature search was limited to include peer-reviewed articles written in English that were published between 2012 and 2022, with the bulk between 2018 and 2022 (last updated on 24 March 2022)

Using a combination of keywords with the "water treatment" category, approximately 760 search results were returned including articles and proceeding papers. Following a refinement for peer-reviewed articles and proceeding papers that focused on chlorination, adsorption, and membrane processes, approximately 400 search results remained. After a text review, 28 references were included for inclusion in this in-depth review (11 for chlorination, 10 for adsorption, and 7 for membrane). The inclusion criteria were based on the scope of the available papers and the availability of specific AI methods and ML models. For the "water systems monitoring" category, approximately 220 search results were returned. Following a similar text review for peer review, scope, and specific AI or ML mention, 12 references that displayed a robust representation of available research were included for this in-depth review. Similarly, the initial searches for "hydroponics" and "aquaponics" combined netted approximately 100 search results (approx. 60 for hydroponics, 40 for aquaponics). Utilizing the refinement criteria, 10 references were selected for inclusion in this in-depth review.

## 3. Machine-Learning Models, Artificial-Intelligence Methods, and Smart Technology

ML models used in water applications are briefly summarized below in Section 3.1. A brief mention of utilized AI methods is also included. A section on smart technologies as defined in this review paper is included in Section 3.2, which are considered the Internet of Things, smart sensors, and systems based on these technologies, and are often integrated with AI/ML models and methods. All of these techniques have been studied for uses in water- and wastewater-treatment processes including chlorination, adsorption and membrane filtration, water-quality management including dissolved oxygen and water level, as well as water-quality-index modeling and/or hydroponics and aquaponics farming.

### 3.1. Machine-Learning Models and Artificial-Intelligence Methods

AI and ML models and methods are briefly summarized below in Table 1. Their general usages, specific usages in water treatment and modeling applications, advantages, and disadvantages are highlighted to aid in the selection of appropriate models and methods for water treatment and monitoring applications. Further peer-reviewed and published textbook sources that supply the necessary foundational and in-depth explanations of these methods and models are also included in the final column. These water treatment and monitoring applications are not intended to be all-encompassing, but to represent the published peer-reviewed journals that were selected based on the methodology explained above. Most of these included ML methods would fit the "black-box" archetype and would be considered a consistent "disadvantage" for many of the models (notably excluding GA/GPs).

**Table 1.** A summary of AI methods and ML models used in water treatment and monitoring.

| Leaning and Modeling Technique | General Applications * | Reviewed Water Treatment and Monitoring Applications | Advantages | Disadvantages |
|---|---|---|---|---|
| Random Forest (RF) | • Supervised machine learning<br>• Regression, Classification<br>[13–18] | • Adsorption process percent removal modeling<br>• Simple and hybrid dissolved-oxygen modeling | • Intuitive model architecture<br>• Capable of handling continuous and categorical inputs-even with missing values/data<br>• Relatively stable with less impact due to noise and outliers<br>• Bagging algorithm reduces overfitting and variance | • Accuracy and robustness determined by the "density" of decision trees<br>• Increases in density result in significant increases in model complexity, model training period, and required computational power |
| Support Vector Machines/Regressions (SVM/SVR) | • Supervised machine learning<br>• Regression, Classification/ Pattern Analysis<br>[19–23] | • Disinfection by-product (DBP) modeling<br>• Membrane-process parameter modeling<br>• Biological oxygen demand (BOD) and chemical oxygen demand (COD) modeling<br>• Dissolved-oxygen modeling of rivers<br>• Aquaponics growth rate modeling<br>• Aquaponics growth stage classification | • Capable of handling high dimensional datasets (i.e., high number of inputs vs. lower number of outputs)<br>• Capable of handling small dataset changes<br>• Functional with both linear and non-linear data | • Kernel selection is initially difficult and time consuming<br>• Modeling requires high computational power making SVM/SVR mostly not suitable for larger datasets<br>• Susceptible to noise in datasets<br>• Relatively long training times |
| Fuzzy Inference System (FIS) | • Artificial intelligence<br>• Decision making, system control<br>[24–28] | • Chlorine dosage set-point control<br>• Hydroponics system and environmental control | • Utilization of fuzzy logic rather than binary logic better models the human experience of decision making<br>• Outputs and decisions are easily interpretable with a well-defined system | • Terminology can be interpreted as confusing without knowledge of fuzzy logic<br>• Applicability dependent on operator-defined parameters and experience-prone to human error |
| Genetic Algorithm/Genetic Programming (GA/GEP) | • Evolutionary, stochastic algorithm<br>• Regression, Classification<br>[29–32] | • DBP formation modeling | • Basic concept easy to understand for most<br>• Multi-objective optimization is possible<br>• Robust to both noisy datasets and local maxima/minima<br>• Functional on discrete, continuous, and mixed datasets | • Implementation is often difficult and time consuming<br>• Requires high computational power<br>• Fitness/objective function and operators difficult to derive |

**Table 1.** *Cont.*

| Leaning and Modeling Technique | General Applications * | Reviewed Water Treatment and Monitoring Applications | Advantages | Disadvantages |
|---|---|---|---|---|
| Artificial Neural Network (ANN)-General | ● (Typically) Supervised machine learning ● Regression, Classification [23,33,34] (Figure 1A) | ● Chlorine dosage/set-point ● DBP formation modeling ● Adsorption process parameter modeling ● Membrane-process parameter modeling ● Dissolved-oxygen-concentration modeling ● Hydroponics system control and classification | ● Capable of handling high dimensional datasets ● Modeling/prediction results obtained in a reasonable amount of time ● Forward propagation capable of cheap and fast computation ● See below for specific ANN model advantages | ● High computational power associated with backward propagation stage ● Some models and architecture themselves are difficult to understand ● See below for specific ANN model disadvantages |
| k-Nearest Neighbor (k-NN) | ● Supervised machine learning ● Classification [23,35,36] | ● Aquaponics growth stage classification | ● Easy to implement with little to no training period ● Capable of handling new data additions | ● Poor performance with large datasets and datasets with high dimensionality ● Susceptible to noise, missing data, and outliers |
| Hammerstein-Wiener (HW) | ● Machine-learning model ● Regression [37–41] | ● Dissolved-oxygen-concentration modeling | ● Capable of modeling dynamic datasets that display static non-linearity ● Static non-linearity can be canceled to apply linear algorithms | ● Particularly complex model that is difficult to understand and implement |
| Radial Basis Function (RBF) Kernel | ● Machine-learning function ● Regression, Classification [23,42–44] | ● DBP formation modeling ● Adsorption process removal efficiency ● Membrane-process parameter modeling | ● Performs faster with less computational power than traditional ANN models ● Less susceptible to local minima/maxima issues ● Capable of handling noisy datasets ● Simple three-layer (input, hidden, output) architecture | ● Complexity greatly increases with increasing neurons in the model's one hidden layer ● Difficulty handling increasingly non-linear datasets |
| Recurrent Neural Network (RNN)/Long Short-Term Memory (LSTM) | ● Supervised machine learning ● Regression, Classification [23,45–48] (Figure 1B) | ● Membrane-process parameter modeling ● Dissolve oxygen concentration modeling | ● Suitable for time-series datasets and modeling ● Suitable for sequential datasets and modeling ● No limit to the length of dataset inputs | ● Requires high computational power ● Requires large and diverse datasets making training difficult |

**Table 1.** *Cont.*

| Leaning and Modeling Technique | General Applications * | Reviewed Water Treatment and Monitoring Applications | Advantages | Disadvantages |
|---|---|---|---|---|
| Convolutional Neural Network (CNN) | ● Supervised machine learning<br>● Regression, Classification, Segmentation [23,49–52] (Figure 1D) | ● DBP formation modeling | ● Results are typically regarded as highly accurate<br>● As the model runs in parallel, results are obtained quickly<br>● Excel at solving with image-based inputs | ● Model and architecture themselves are extensive and complicated<br>● Requires high computational power |
| Adaptive Neuro-Fuzzy Inference Systems (ANFIS) | ● Supervised machine learning<br>● Regression, Classification [53–55](Figure 1C) | ● DBP formation modeling<br>● Adsorption process removal efficiency modeling<br>● Membrane-process parameters modeling<br>● Dissolved-oxygen-concentration modeling<br>● BOD/COD modeling | ● Combined important advantages of ANN models with the FIS including:<br>● No need to rely solely on the human experience as a FIS does<br>● Relatively fast learning<br>● Uses both numerical and linguistic language during modeling<br>● Capable of handling non-linear datasets<br>● Able to classify and recognize patterns as an ANN model does | ● Requires high computational power that increases with the number of rules implemented<br>● Highly susceptible to performance issues with smaller datasets; more so than other ANN models<br>● Membership function type and number are vital and can be difficult to implement to create acceptable accuracy |
| Extreme Learning Machine (ELM) | ● Supervised machine learning<br>● Regression, Classification [56–58] | ● Dissolved-oxygen-concentration modeling | ● Relatively short training times<br>● Suitable for pattern classifications | ● Often faces over-fitting or under-fitting if too many/few hidden nodes are utilized |

* The references cited in "General Applications" are foundational source material of the AI models.

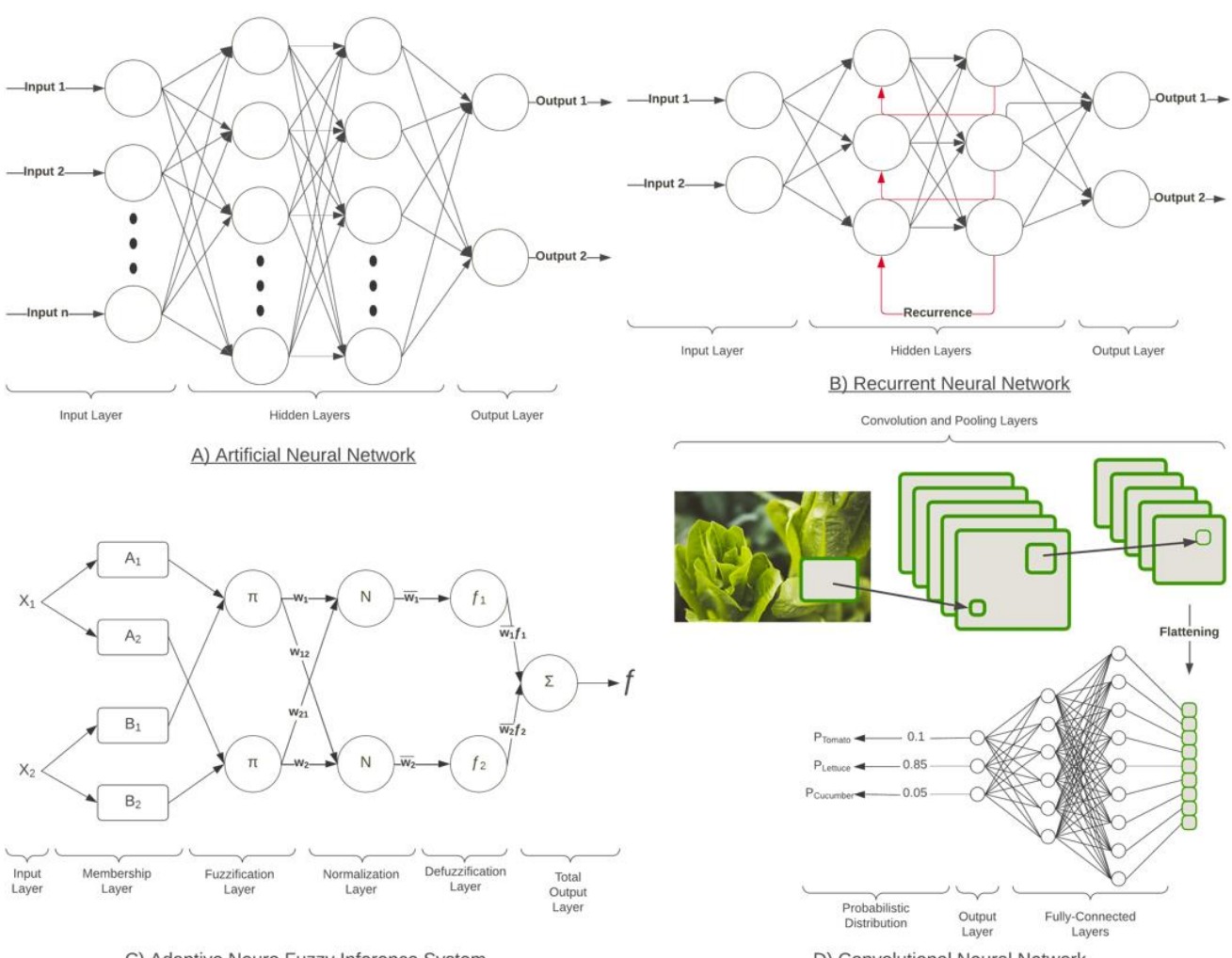

**Figure 1.** Four Common Neural-Network Structures in Water Treatment and Monitoring. (**A**) Four Layer Artificial Neural Network (**B**) Four Layer Recurrent Neural Network (**C**) Typical Adaptive Neuro Fuzzy Inference System Structure (**D**) Typical Convolutional Neural Network Structure.

### 3.2. Smart Technology—The Internet of Things (IoT) and Smart Sensing Technology

The Internet of Things is a descriptor for a network of physical objects that can connect to the internet (or other communication networks) and that are often endowed with some sort of analytical process (such as environmental sensing) using software, hardware or other technologies. In the context of this paper, the IoT in water applications often includes internet-enabled systems equipped with pressure sensors, flow sensors, and/or water-quality/characteristic sensors [59]. The intent is typically to exchange data with other connected devices or networks over the life and duration of the sensor or other technology [60], often for system optimization, transparency or ease of use [61]. The IoT creates a cooperative network of data collection that can be stored locally or offsite without a human ever physically needing to take the data themselves or operate the physical object. As such, the long-lasting function and life of the connected device must be maintained. Though not technically artificial intelligence, the IoT can be fused with AI to create what has been coined as the "Artificial Intelligence of Things", which would marry this data-collection process to feed AI with critical inputs for its learning process [62].

Smart sensing technology can be related to IoT, but often represents a broader breadth of systems that do not need to be defined by their collectiveness and can also include stand-alone or isolated systems/sensors. To achieve the designation of "smart" sensing technology, the sensors must have some function beyond their general sensing abilities [63],

which is generally achieved through an actionable decision or automation. For example, a thermostat that both measures the temperature of a room and interfaces with a furnace in order to achieve a set temperature, thus not inherently requiring a connection with other smart devices in the home. The smart designation can be enhanced through the ability to wirelessly interact with other systems through Wi-Fi or Bluetooth capabilities.

## 4. Applications in Water and Wastewater Treatment

Artificial intelligence and machine-learning techniques have been studied in several water- and wastewater-treatment applications. This section will serve as a cross-section of three common treatment processes employed at water- and wastewater-treatment plants. Much of the input data utilized by the reviewed journals were collected and disseminated by treatment-plant staff or other regulatory bodies, relying on traditional collection methods. Using smart-technology integration with the reviewed AI methods or ML models, the burden of data collection can be decreased. More data is also likely to increase the accuracy of selected ML models. Ultimately, this is not intended to represent the gamut of research into AI and ML application in the water-treatment industry but rather a representation of current research interest.

AI methods have been demonstrated to be effective in controlling chlorination, while ML models are effective in modeling DBP concentrations, as well as modeling important parameters for adsorption and membrane-filtration processes. The results are often evaluated using various statistical measures including the coefficient of correlation (R), the coefficient of determination ($R^2$), the mean average error (MAE), the mean square error (MSE), the root mean square error (RMSE), and relative error (RE).

### 4.1. Chlorination and Disinfection By-Product Management

Disinfection in a water- and wastewater-treatment plant is the process by which microorganisms and viruses are killed or inactivated, mainly with chlorine-based disinfectants [64]. While chlorination is effective as a disinfectant, it also poses human health hazards [65]. Beyond its ability to cause acute toxicity in humans, chlorine is also known to interact with bromide and organic matter naturally found in water systems to form what is known as disinfection by-products. Disinfection by-products (DBPs) are suspected human carcinogens and reproductive disruptors, and have received increased scrutiny from regulators all over the world [66]. DBPs mainly belong to two larger subcategories: trihalomethanes (THMs) and haloacetic acids (HAAs). THMs are regarded as the most common form of DBPs as their formation is associated with chlorine disinfectants [67]. Haloacetic acids are commonly tested for five or nine common haloacetic acids and are commonly referred to as HAA5 or HAA9. The entire mechanism behind the formation of DBPs in drinking water is not known, making their prediction and mitigation an ideal candidate for ML technologies. When learning has been achieved, mitigation through control using AI methods is possible.

The published applications using AI methods to control chlorination and ML to model and predict DBP formation/chlorine requirements are presented in Table 2. Many researchers performed model testing on surface waters that undergo treatment at drinking-water plants utilizing chlorine as the primary disinfectant, though some studies did involve pre-chlorination peroxide/ozonation. Researchers also noted success in modeling DBP concentrations in the treated water-distribution networks, and directly at consumer homes and taps. Common model inputs include water temperature, pH, chlorine concentration, contact time, and TOC/DOC concentrations. Other successful models have implemented inputs using bromine concentration, $UV_{254}$, algae concentrations, chlorophyll-a concentrations, and DBP-precursor chemical markers.

**Table 2.** AI methods to automate and ML to model and predict DBP formation/chlorine requirements.

| Target Compound (s) | Water Source | Disinfectant | AI/ML Technique Used | Input Variables | Output | Reference |
|---|---|---|---|---|---|---|
| Chlorine dose and free residual chlorine (FRC) set point | Surface water | Chlorine | ANN | Reservoir set-point output, FRC of treated water tank, FRC output of WTP (mg/L), WTP production flow rate, compensating system flow rate, dosage error | Chlorine dosage, WTP FRC set point | [68] |
| Chlorine dose | Surface water | Chlorine (ClO) | FIS | Raw water total organic carbon (TOC), pH, contact time, temperature | Chlorine dosage recommendation, FRC | [69] |
| Total tri-halomethanes (TTHMs) | Surface water | Chlorine | SVM, ANN, GEP | pH, temperature, contact time, $Cl_2$/DOC, bromine concentration | TTHM effluent concentration | [70] |
| TTHM | Surface water | Chlorine ($Cl_2$) | SVM, ANN | pH, temperature, residual chlorine, TOC, $UV_{254}$ | TTHM effluent concentration pre-monsoon season (PrM) and post-monsoon season (PoM) | [71] |
| TTHM | Surface water | Chlorine | ANN | Temperature, pH, TOC, algae concentration, chlorophyll-a concentration, pre, middle, and post chlorine concentration, total chlorine concentration | TTHM effluent concentration | [72] |
| DCAA, TCAA, BCAA, HAA5, HAA9 | Tap water | Chlorine | RBF-ANN | Dissolved organic carbon (DOC), $UVA_{254}$, bromine concentration, temperature, pH, Cl2 concentration, NO2-N concentration, NH4+-N concentration | DBP tap concentration | [73] |
| TTHM, TCM, BDCM | Tap water | Chlorine | RBF-ANN | pH, temperature, $UVA_{254}$, $Cl_2$ concentration | DBP tap concentration | [74] |
| TTHM, TCM, BDCM, THAA, DCAA, TCAA | Surface water | Peroxide/Ozone Chlorine | ANN, CNN | Fluorescence spectra | DBP effluent concentration | [75] |
| TTHMs, TCM, BDCM, DBCM | Surface water | Chlorine | ANFIS | Temperature, pH, $UVA_{254}$, $Cl_2$ concentration, dissolved-organic-carbon concentration | DBP effluent concentration | [76] |
| DCAA, TCAA | Lab-created | Chlorine | ANN, RF, SVM | Number of aromatic bonds, hydrophilicity, electrotopological descriptors related to electrostatic interactions, and atomic distribution of electronegativity | DBP effluent concentration | [77] |

The most tested ML model used for chlorination and DBP prediction is the ANN, with other applications involving support vector machines, fuzzy inference systems, and genetic algorithms. In comparative studies, ANNs typically outperformed both GAs and

SVMs, though there are some cases of SVMs providing a slight advantage when $R^2$ is used as a point of comparison [60,67]. Common DBPs that were modeled and/or predicted include total trihalomethanes (TTHM) and total haloacetic acids (THAA), with some studies focusing on specific DBP compounds including dichloroacetic acid (DCAA), trichloroacetic acid (TCAA), bromochloroacetic acid (BCAA), HAA5, HAA9, trichloromethane (TCM), bromodichloromethane (BDCM) and dibromochloromethane (DBCM). Predictions for TTHMs or THAAs versus their compounds did not differ widely in statistical model-validation numbers.

### 4.2. Adsorption Processes

Adsorption processes are generally regarded as both a physical and chemical treatment option for removing a wide range of contaminants and pollutants in both the water-treatment and wastewater-treatment industries. The process of adsorption involves an exothermic mass-transfer surface process that causes the transfer of some target molecule (or adsorbate) from a fluid to a solid surface (or adsorbent, but also often referred to as the sorptive media in the industry) [78]. Due to the complexity of the interactions that affect the efficacy of an adsorptive process [79], it is often difficult for plants to precisely calculate the important parameters and ultimate removals of the adsorption process. Reducing this complexity would enable a treatment plant to extend a sorptive media's life and increase a treatment plant's effectiveness and confidence that it is effectively treating the water according to any applicable rules and regulations. To further optimize the process, researchers have identified models using ML to make important predictions for the adsorption process. ML for adsorption processes have the potential to support operator decisions.

The published applications using ML to model and predict adsorption-process parameters are presented in Table 3. Studies have been published modeling adsorption processes with water streams contaminated with metals, industrial dyes, and organic compounds. Adsorbent media ranges widely and includes carbonaceous materials and metal-based nanocomposites, among others. Inputs commonly used during ML modeling of adsorption processes include pH, water temperature, adsorbent dose, contact time, and initial adsorbate concentration. Individual models have included inputs utilizing adsorbent particle size, system flow rate, agitation speed, bed height, and BET surface area, among others. Studies that included various organic pollutants operated with varying compound-specific parameters such as target-contaminant molar mass. The majority of the published studies that are included in this review pertained to models with outputs of adsorbate percentage removal (also known as adsorption efficiency), though some models sought to predict adsorption capacity, non-dimensional effluent concentrations, and the relative importance of input water-quality parameters.

**Table 3.** ML to model and predict adsorption processes and removal rates.

| Adsorbate | Adsorbent | ML Technique Used | Input Variables | Output | Reference |
|---|---|---|---|---|---|
| Copper ions | Attapulgite clay | RF, ANN, SVM | Initial copper concentration, adsorbent dose, pH, contact time, addition of NaNO3 | Adsorbate percent removal | [80] |
| Asphaltenes | Nickle(II) Oxide Nanocomposites | RBF-ANN, ANN, SVM, | Type of nanocomposite, pH, amount of adsorbent over adsorbate concentration, temperature | Adsorbate percent removal | [81] |
| Various organic pollutants | Activated carbon | ANN, SVM, ANFIS | Molar mass of target contaminant, initial concentration, flow rate, bed height, specific surface area, contact time | Non-dimensional effluent concentration | [82] |
| As (III) | Various | ANFIS | Initial concentration, adsorbent dose, pH, contact time, agitation speed, temperature, solution volume, inoculum size, flow rate | Adsorbate percent removal | [83] |
| Methylene blue (MB), Cd(II) | Natural walnut activated carbon | ANN | pH, adsorbent mass, MB concentration, Cd(II) concentration, contact time | Adsorbate percent removal | [84] |
| Sunset yellow (SY) | Neodymium modified carbon | ANN | Adsorbent dose, initial concentration, contact time | Adsorbate percent removal | [85] |
| Ni(II), Cd(II) | Typha domingensis (Cattail) biomass | ANFIS | pH, adsorbent dosage, metal-ions concentration, contact, biosorbent particle size | Input parameters influence on removal efficiency | [86] |
| Zn(II) | Rice husk | ANN | Initial concentration, contact time, temperature | Adsorption capacity | [87] |
| Phosphate | Encapsulated nanoscale zero-valent iron | ANN | pH, phosphate concentration, adsorbent dose, stirring rate, reaction time | Adsorbate percent removal | [88] |
| Various organic pollutants | Activated carbon | ANN | Molar mass of target contaminant, initial concentration, flow rate, bed height, particle diameter, BET surface area, average pore diameter | Non-dimensional effluent concentration | [89] |

For studies including metal, organic, and industrial-dye contaminants, the ANN was the most used ML model. Other models that researchers studied with notable success include ANFIS, SVM, and RF. On average, ANN, SVM, and RF ML models performed adequately, generally achieving $R^2$ values greater than 0.9, and in some cases, greater than 0.99 [80,81]. In most cases, SVM models performed slightly better than ANN models, producing both $R^2$ and RMSE values of better statistical value. In one case, the optimized ANFIS model performed poorly in comparison to other success models for adsorption processes, achieving an R = 0.813, and was noted as the worst performing in a comparison between ANN, ANFIS, and SVM models [82], though in another it achieved the adequate performance with an $R^2$ = 0.9333 [83].

### 4.3. Membrane-Filtration Processes

Membrane processes in water and wastewater treatment refer to the separation of contaminants using a barrier or filter. The water is passed through the membrane usually due to high-pressure differentials between one side of the membrane and the other side [90]. The smaller the pore size, the more pressure is required to pass the water through the membrane. Membrane processes are typically used for contaminants that are difficult or costly to remove by chemical or physical means, but also for contaminants that require a high level of removal that simply cannot be achieved by other chemical or physical means [91]. The most used membrane processes are microfiltration, ultrafiltration, nanofiltration, and reverse osmosis.

Applications using ML to optimize, model, and predict the membrane-filtration process are presented in Table 4. Researchers have created models that function with microfiltration, ultrafiltration, nanofiltration, and reverse osmosis. A study involving a submerged membrane bioreactor has also been included in this review. Water sources tested using these models include a wide array of pollutants and natural compounds, including petroleum/oil, natural organic matter, various industrial and pharmaceutical wastes, and simple salt/ocean water. Similar to previous sections involving ML in water/wastewater-treatment applications, ANNs are the dominant model used. Other models that have been utilized for membrane-filtration-process modeling include ANFIS, SVM, and specific forms of ANNs including RNNs (some of which utilize LSTM).

**Table 4.** ML to model and predict membrane-filtration parameters.

| Membrane Type | Water Source | ML Technique Used | Input Variables | Output | Reference |
|---|---|---|---|---|---|
| Titanium-based ceramic ultrafiltration | Petroleum production wastewater | ANN, ANFIS, RBF-ANN | pH, temperature, time, transmembrane pressure, crossflow velocity | Permeate flux | [92] |
| Nanocomposite ultrafiltration | Various | ANN | Polymer type, polymer concentration, filler concentration, filler concentration, average filler size, solvent type, solvent concentration, contact angle | Solute rejection (SR), pure water flux (PWF), flux recovery (FR) | [93] |
| Microfiltration | Dilute suspension mixture of crude oil, dilute suspension mixture of tween-20 | ANN | Flux rate, filtration time, shear rate | Transmembrane pressure (TMP) | [94] |
| Submerged membrane bioreactor | Palm oil mill effluent | RNN | Pump voltage, airflow, transmembrane pressure OR flux | TMP, permeate flux (PF) | [95] |
| Reverse osmosis | Saltwater | ANN | Membrane operating period, the time between cleanings, water temperature, input concentration, inflow, inlet pressure, recovery percent | Pressure drop (PD), salt passage (SP) | [96] |
| Nanofiltration | Surface water w/ natural organic matter | RNN (LSTM) | Fluorescence regional integration, pressure, initial flux, DOC concentration, fouling layer thickness | Permeate flux (PF), fouling layer thickness (FLT) | [97] |
| Nanofiltration/reverse osmosis | Pharmaceutical wastewater | ANN, SVM | Effective diameter of the target compound, logD, dipole moment, molecular length, molecular equivalent width, molecular weight cutoff, sodium chloride salt rejection, zeta potential, contact angle, pH, pressure, temperature, recovery | Rejection percentage of the target compound | [98] |

ML techniques for modeling membrane-filtration processes seek to output several different variables, commonly including transmembrane pressure, permeate flux, and solute rejection. Inputs that exist in some of these published studies include pH, temperature, contact/filtration time, transmembrane pressure, and flux rate, among many, and more specific, options. Again, due to the range of models testing for different parameters, it is difficult to fully compare the statistical values that many of these studies obtained. Ultimately, ANN, RNN, and SVM models performed adequately well in terms of their respective $R^2$ values, consistently achieving values greater than 0.9, and in many cases, achieving values greater than 0.99 [92,94,98] (Table 4).

### 4.4. Artificial Intelligence in Water Treatment: A Brief Case Study of ANN, SVM, and RF Models for Adsorption-Efficiency Prediction

The published study conducted by Bhaget et al. ([80]) focused on the prediction of copper-ion removal in an adsorption process relying on attapulgite clay as the primary adsorbent. Three models were developed and compared to determine the optimal form of prediction. These models included an artificial neural network, a support vector machine, and a grid-optimization-based random forest. This study conducted by Bhaget et al. was selected due to their in-depth discussion of the construction of their model and their methods for input selection. This ultimately aids in furthering the development of these intelligent models and methods by allowing future researchers to better understand the processes and details of Bhaget et al.'s implementation.

The model inputs for all three intelligent options included initial copper concentration, adsorbent dosage, pH of the water, contact time, and the ionic strength of the solution (upon the addition of $NaNO_3$). Copper-ion concentration was held at a constant level of 50 mg/L to determine the effect the other input variables had on the adsorption efficiency. The adsorbent dosage varied from 2 to 15 g/L, pH varied from 2.0 to 12.0, $NaNO_3$ concentration varied from 0 to 0.5 mol/L, and contact time was evaluated varying from 5 to 120 min.

The models were all developed using open-source software for the programming language R, known for its usage in statistical computation. For training and testing, the dataset was divided into two with 80% used for training and the remaining 20% used for testing. The ANN model consisted of a four-layer model (one input layer, two hidden layers, and one output layer) where the first hidden layer contained five nodes, while the second hidden layer contained three nodes. Each neuron relied on a linear output function. The SVM model was developed using a linear kernel. Finally, the RF model utilized 76 samples to develop decision trees relying on the bootstrapping method (which divides data into several subsets through random replacement, allowing each decision tree in a forest to have its "own" random subset for training purposes [99]).

Tests were run on each model using a varying set of inputs. The input sets included: (1) initial copper concentration; (2) initial copper concentration and adsorbent dosage; (3) initial copper concentration, adsorbent dosage, and contact time; (4) initial copper concentration, adsorbent dosage, contact time, and pH; (5) initial copper concentration, adsorbent dosage, contact time, pH, and $NaNO_3$ concentration. The models were noted to perform best using all five inputs. Ultimately the RF and ANN models were determined to display the best performance in terms of accuracy, achieving correlation coefficients greater than 0.99, while SVM achieved a maximum correlation coefficient of 0.93.

## 5. Applications in Water-Quality Management

Artificial intelligence and machine-learning techniques have been studied in water-quality management. This section will serve as a cross-section of some water-quality-management models including dissolved oxygen, among other water-quality parameters and indices, and river-water-level monitoring.

ML models have been demonstrated to be useful for the prediction and modeling of water-quality-management parameters. The results were commonly evaluated using various statistical measures, potentially including the coefficient of determination ($R^2$), the

mean square error (MSE), the root mean square error (RMSE), the normalized root mean square error (NRMSE), the mean absolute percentage error (MAPE), the Nash–Sutcliffe efficiency coefficient (NSE), the Pearson correlation coefficient (PCC) and/or accuracy (ACC).

*5.1. Water-Quality Management*

Water-quality management is an important task necessary for the health and good function of aquatic ecosystems. Often, human activity can hurt the water quality of rivers and other waterways, and tracking this effect is vital to maintaining these ecosystems. A commonly tracked parameter used to discern the health of a river or other waterway is the dissolved-oxygen concentration. Hypoxia (or the lack of dissolved oxygen in waterways) is becoming increasingly prevalent, generally because of increased nutrient loading and global warming [100].

Due to the interactions between dissolved-oxygen (DO) concentrations and human activity/pollution, it is increasingly important to measure DO as a means of predicting, and possibly preventing, hypoxic zones from dealing widespread damage to these aquatic ecosystems. Accurate and real-time results are often most favorable as moderate decreases in DO represent potentially fatal results in certain species [101]. In some cases, DO sensing can be obfuscated by environmental factors, demonstrating a present need for models and methods that can overcome the traditional sensing methods' shortcomings [102].

Table 5 presents some studies on modeling water-quality-management parameters with ML, chiefly including dissolved-oxygen modeling, with additional studies focusing on the more general water-quality index, and/or future water levels for rivers. Locations used for sensing and monitoring water-quality parameters mainly include rivers, with one WWTP included for BOD and COD monitoring.

Common inputs for water-quality modeling using ML include pH, water temperature, and BOD levels. These inputs are also generally the same for water-quality-index (WQI) monitoring and BOD/COD modeling, with the inclusion of dissolved oxygen as an input in the case of the published study for WQI included in this review, while water-level monitoring relies exclusively on past water levels and robust training data.

While studies modeling aides for water-quality management using ML techniques mainly utilize ANNs, a wide array of other methods have also been studied including ANFIS, RNN, EML, RT, SVM, HW, and hybrid ML models utilizing some of them with RF models. Most studies' models demonstrated accurate predictions, but this is ultimately location dependent. On average, ANFIS models outperformed typical ANN and SVM models in almost all the published studies reviewed here and presented in Table 5, and in some cases were outperformed by hybrid models. Water-level forecasts were accurately predicted using both ANN and ANFIS models, achieving $R^2$ values greater than 0.999 with both models.

**Table 5.** ML to model and predict water-quality parameters and environmental variables.

| Location | ML Technique Used | Input Variables | Output | Ref |
|---|---|---|---|---|
| Mathura, India (Yamuna River) | ANN, ANFIS | pH, BOD, water temperature, dissolved oxygen (DO) (all inputs have independent variables taken at stations upstream, midstream, and downstream) | Upstream, midstream, and downstream DO concentration | [103] |
| Oregon, USA (Link & Klamath Rivers) | ELM, ANN, ANFIS, RF | Hourly temperature, pH, specific conductivity | DO concentration | [104] |
| Malaysia (Kinta River) | ANN, RNN (LSTM), ELM, HW, ANN-RF, RNN (LSTM)-RF, ELM-RF, HW-RF | BOD, COD, pH, NH3, temperature, chlorine, calcium, sodium, and total solids concentrations | DO concentration | [105] |
| Tabriz, Iran (Tabriz WWTP) | ANN, ANFIS, SVM | Daily influent BOD/COD, TSS, pH, previous BOD/COD effluent | BOD, COD effluent | [106] |
| Kedah, Malaysia (Muda River) | ANN, ANFIS | Water level for $(t-1),(t-2)$, and $(t-3)$, where $t-1$ is the water level 1 h ago, and so on | Future water level (in one hour) | [107] |
| Palla, India (Yamuna River) | ANN, ANFIS | DO, pH, BOD, NH4, water temperature | Water-quality index | [108] |
| Nizamuddin, India (Yamuna River) | ANN, ANFIS, SVM | pH, BOD, COD, flow rate, NH3 concentration, water temperature | DO concentration | [109] |
| Udi, India (Yamuna River) | ANN, ANFIS, SVM | pH, BOD, COD, flow rate, NH3 concentration, water temperature | DO concentration | [109] |
| Kelantan, Malaysia (Kelantan River) | ANN | DO concentration, BOD, COD, pH, ammonia nitrogen concentration, suspended solids | DO concentration, BOD, COD, pH, ammonia nitrogen concentration ($NH_3$-NL), suspended solids (SS) | [110] |
| Hilo, Hawaii (Wailuku River) | ANN, ELM, SVR | Hourly turbidity, hourly salinity, hourly water temperature, hourly river flow | Turbidity w/ river flow at t | [111] |
| Mesa, Arizona (algae cultivation pond) | RNN (LSTM) | Microbial potentiometric sensor measured open-circuit potentials | Blue-green algae conc., conductivity, chlorophyll conc., DO, pH, turbidity | [112] |
| Johor State, Malaysia (Johor River) | ANFIS | Temperature, conductivity, salinity, nitrate, turbidity, phosphate, chloride, potassium, sodium, magnesium, iron, and *E-coli* concentrations | Suspended solids, pH, ammoniacal nitrogen | [113] |
| Thailand (Chao Phraya River) | SVM with varying kernel functions | BOD, DO, fecal coliform bacteria, total coliform bacteria, ammonia concentration, salinity | ammonia concentration, total coliform bacteria, fecal coliform bacteria, BOD, DO, salinity | [114] |

*5.2. Artificial Intelligence in Water-Quality Management: A Brief Case Study of ANFIS and ANN Models for WQI Prediction*

ML-based models have been developed to predict the water-quality index for the river Ganga and its tributaries ([108]). Both models (ANN model and ANFIS model) relied on inputs of dissolved-oxygen concentration, pH, BOD, ammonium nitrate concentration,

and water temperature. This case study will highlight the methods used to achieve the AI models to better understand their construction and relative accuracy compared to one another. The study conducted by Gaya et al. was selected due to their inclusion of the ML models' structure, along with their decision to utilize similar ML models with varying structures to test the effect varying inputs and hidden layers had on model accuracy.

Both the ANN model and the ANFIS model were developed using programming packages within MATLAB R2017b. Both models utilized a dataset of which 70% was employed for training (referred to as calibration) and 30% was used for testing (referred to as validation). The ANN model was a simple three-layer neural network (one input layer, one hidden layer, and one output layer). Input data were normalized from 0 to 1 before being fed to the ANN model. As is commonly used with these models, the ANN model relied on backpropagation, meaning that input training data are fed through the model, passing through the output layer, where training error is propagated backward until the selected level of accuracy is achieved. The models were tested using five different structures that differed for the ANN and ANFIS model. For the ANN model, the structure was varied and included varying numbers of nodes. ANN model 1 (ANN-1) used one input node of dissolved oxygen, one hidden-layer node, and one output node. ANN-2 used two input nodes of dissolved oxygen and pH, two hidden-layer nodes, and one output node. ANN-3 used the previous plus BOD, three hidden-layer nodes, and one output node. ANN-4 used the previous plus ammonium nitrate concentration, four hidden-layer nodes, and one output node. ANN-5 used the previous water temperature, six hidden-layer nodes, and one output node. The best performing ANN model was noted as ANN-2.

The ANFIS models were tested using five different structures with variable input variables and two triangular membership function inputs with constant output. ANFIS model 1 (ANFIS-1) utilized all five input variables. ANFIS-2 utilized four input variables, ANFIS-3 utilized three input variables, ANFIS-4 utilized two input variables, and ANFIS-5 utilized two input variables. Interestingly, the best-performing model was noted as ANFIS-2, but variable combinations were not as readily available for ANFIS models as they were for ANN models. Both ANN and ANFIS models were noted for their relative success in predicting actual WQIs, achieving high determination coefficients greater than 0.99.

## 6. Applications in Water-Based Agriculture

Smart technology in conjunction with artificial intelligence and machine-learning methods has garnered interest in some sectors of the research community. This section will serve as a cross-section of two water-based agricultural methods: hydroponics and aquaponics. Smart technology both coupled with and independent of AI methods and ML models (referred to below as "Smart Systems") has been demonstrated to be effective in automating and monitoring the growth process and health of these water-based agricultural systems. The results are evaluated using various statistical methods including the system accuracy, the coefficient of determination ($R^2$), the mean average error (MAE), the false-positive rate (FPR), and the system error (Err).

### 6.1. Hydroponics and Aquaponics

Hydroponic farming and hydroponic systems are methods of plant cultivation that do not use soil. Plants are grown in (an often specifically tailored) nutrient solution that provides the plant with all its nutrient and water needs. While this is a far more technical form of cultivation compared to traditional farming, hydroponics has the distinct advantage of producing higher crop yields, with greater plant density in significantly less space and with lower average water usage [115]. Crops are grown suspended in a tailored nutrient solution that must also be kept at the proper pH for growing, and the growing rooms must be kept at the proper humidity and temperature [116]. Nutrient solutions are typically stored in separate tanks and are delivered to the crops utilizing a pump and pipe network.

Aquaponics is like hydroponics and is often considered a subset of hydroponic farming. Plants are still typically grown without the use of soils, but instead of relying on a tailored

solution for nutrients, a more sustainable cycle is employed [117]. In aquaponic systems, plants receive their nutrients from the by-products of fish (typically fecal matter) stored in adjacent (or near-adjacent) tanks and connected through a pump and pipe network. In return, the crops often act as water purifiers for the fish through the removal of their by-products and fecal matter [118]. It is often a difficult process for cultivators to maintain and optimize hydroponic and aquaponic setups. Thus, researchers have been looking into artificial-intelligence models and have been more commonly using smart technology to help ease some of the burdens of controlling a hydro/aquaponics system.

For the purposes of this paper, AI methods, ML models, and smart applications used in the hydro and aquaponic studies are presented in Table 6 and referred to as "Intelligent Models, Methods and Technology". In contrast to other sections presented in this review, many of the applications involve control and monitoring using smart and/or internet-enabled devices, often using the IoT. Some of the included articles do not rely on a proper IoT, and instead use smart sensors (that may have internet functionality) for control and automation. Like more traditional forms of AI and ML, these setups rely on the use of critical inputs for information monitoring and system action. Common aqua and hydroponic sensor data include pH, water temperature, air temperature, humidity, nutrient/plant height (often measured using an ultrasonic sensor), and electrical conductivity (which is meant to be analogous to nutrient loading). Other aquaponic-specific inputs include total dissolved solids and ammonia concentration.

In studies where IoT or smart sensing is utilized for monitoring system health and automation, outputs are commonly related to nutrient-pump feeding, humidity, temperature controls, pH control, and light control. To achieve these levels of automation, these IoT systems can be paired with AI methods and ML models, such as FIS or ANN. These systems are also paired with central control/processing units (though due to the scale of many of these studies, they are often technically considered microprocessors).

Arduino-based controllers are the most popular among the reviewed applicable published papers. Studies indicate that there has also been some success in implementing ML models for crop harvestability, crop height, fish weight, and nutrient solution constituent concentrations. These outcomes have been achieved using k-NN, SVM, and ANN. Studies integrating the models and smart technology have noted positive plant growth compared to traditional methods and have allowed operators to employ remote-monitoring techniques.

**Table 6.** Intelligent models, methods and technology to monitor and predict plant growth in hydroponic and aquaponic agriculture systems.

| Type of Water-Based Agriculture | Intelligent Models, Methods, and Technology Utilized | Input Variables | Output | Reference |
|---|---|---|---|---|
| Aquaponics | k-NN, SVM | Images of crops | Growth stage classification (vegetative, head development, and harvestable) | [119] |
| Aquaponics | IoT | Digital light, water level/plant height (ultrasonic), air temperature and humidity, water temperature (in a fish tank), electrical conductivity, pH | System health notifications, activation/deactivation of actuators for fish feeding, water heating, and grow lights | [120] |
| Aquaponics | Smart Sensing and Control | Humidity, water temperature, pH, light intensity, total dissolved solids, room temperature, flow between systems | Water pump (for circulation), air pump (for water oxygenation), grow light, fish feeder | [121] |

**Table 6.** *Cont.*

| Type of Water-Based Agriculture | Intelligent Models, Methods, and Technology Utilized | Input Variables | Output | Reference |
|---|---|---|---|---|
| Aquaponics | SVR | Water temperature, ambient temperature, pH, amount of fish feed used, TDS, ΔpH, Δwater temperature, Δambient temperature, Δfish weight, Δplant height | Fish growth rate, plant growth rate | [121] |
| Aquaponics | IoT, FIS | Temperature, turbidity, pH, dissolved oxygen, TDS, ammonia concentration, water level | System health notifications, temperature, and ammonia control using FIS | [122] |
| Hydroponics | IoT, ANN | pH, temperature, light intensity, humidity, water level | Water pump (for nutrient or pH control), light control, humidity control. | [123] |
| Hydroponics | Novel AI monitoring system, Smart Sensing and Control | Humidity, temperature, pH, water level | System health notifications | [124] |
| Hydroponics | FIS | pH, humidity | pH control, humidity control | [125] |
| Hydroponics | Self-Learning AI | UV-vis spectroscopy | Total nitrogen, total phosphorus, total potassium, pH | [126] |
| Hydroponics | IoT | Room temperature, room humidity, water temperature, water pH, horticultural lighting, fertilizer level | System health notifications | [127] |

*6.2. Smart Technology and Artificial Intelligence in Water-Based Agriculture: A Brief Case Study of IoT and FIS in Aquaponics*

The published study conducted by Rozie et al. ([122]) implemented the IoT and an FIS using internet-connected sensors and data-connected rules and membership. The study conducted by Rozie et al. was included for its in-depth explanation and inclusion of the necessary sensors and tools to create an IoT network and a functioning automatic system using an AI method. Their clear explanation of the membership functions used to create their FIS is also an important reason why this paper was selected. Ultimately, these inclusions and explanations will aid future researchers in reproducing the results presented below while also maintaining confidence in the knowledge that the system was sufficiently explained.

The IoT-based sensors captured data relating to pH, temperature, turbidity, dissolved oxygen, total dissolved solids, ammonia concentration, and water level while relying on them for a cloud-based storage system. Data were stored in various formats (CSV, excel, pdf, and images) with the intention of being used for creating a timeseries for the study and long-term observation of aquaponics processes and variables. Two of the aforementioned datasets, namely temperature and ammonia concentration, were additionally utilized for automation and control using the FIS.

Fuzzification of the data for the FIS was achieved using three membership functions: namely a temperature function with defined "cold", "good", "warm" and "hot" variables, an ammonia function with defined "safe", "warning", and "toxic" variables, and a motor-speed function with "slow", "normal" and "quick" variables. The temperature and ammonia-concentration variables were defined using experience and knowledge derived from the Indonesian National Standards Agency. The motor-speed function was used to control the temperature and ammonia concentration by introducing clean and appropriate temperature water to the system. For example, when the membership function for temperature indicates "hot" and the membership function for ammonia concentration indicates

"toxic", the input for the motor-speed membership function would result in an output of "quick", thereby quickly introducing freshwater to the system to cool water temperatures and dilute ammonia levels. All data were simultaneously recorded and uploaded to a web-application-based user interface and a TelegramBot live-chat feature to aid in remote control and monitoring of the system.

Utilizing the FIS, Rozie et al. were able to achieve acceptable temperature cooling in approximately 2 h versus the 3.5 h that a non-controlled system would naturally take to cool using fixed motor speeds. Ammonia control was also achieved and able to decrease ammonia levels from 0.19 ppm at its peak to approximately 0.04 ppm without direct human intervention. System updates directly to human monitors for other variable inputs recorded were sent to users within 3 s of appreciable changes.

## 7. Common Challenges with AI and ML Implementation in Water Treatment and Monitoring

AI methods and ML modeling have some distinct and advantageous uses compared to traditional models and knowledge. While these successes are important to note, it is also important to note that there are some unique challenges associated with this space that must be discussed for widespread adoption to become possible in water-based applications.

### 7.1. Learning and Reproducibility Challenges

Many artificial-intelligence and machine-learning techniques are subject to poor reproducibility as they are often developed using random weights and tailored activation functions that may only work with data that have similar characteristics as the dataset with which it was originally trained and tested [128]. This not only makes it difficult for organizations to apply a singular AI method or ML model to many different areas of their industry, but it also makes it difficult for researchers to recreate and verify previous methods and conclusions. It is often the case that the specific code used to create their ML models is not included in published works, which instead opt to include general descriptions of their process. While researchers have many reasons to exclude the exact code from their published work (such as it being proprietary), this does very little to boost public confidence, knowledge of their systems, and verifiability, relegating AI and ML further to their "black-box" delegations. This exclusion of code is true for many of the studies reviewed here in this paper.

### 7.2. Data Challenges

Another important challenge to note is that these models are extensively dependent on the selected data. The learning/training process is often the most important step in developing a successful model, and selecting an appropriate dataset is no small task of that process. Due to this, small deviations in natural or treatment processes that were not encountered during the training process can cause a previously successful model to become essentially defunct and unable to process the new information that it was supplied as it was never prepared to handle it. This problem can be further exacerbated by the fact that many real-life systems lack the extensive datasets needed to power many of these ML models. While researchers and academics can select their datasets, many communities that would benefit from these intelligent systems simply do not have the data-management/storage capabilities necessary for the function of these models [129]. So, while larger municipalities and operations may be able to overcome these data-management issues, smaller populations will be left with inadequate information.

### 7.3. Benchmark and Standardization Challenges

This challenge regarding the selection of data also extends into an issue of specificity. As data are often carefully selected for use under specific conditions and locations, it is increasingly difficult to extend existing and proven models to new locations. Ultimately, this makes regulatory adoption significantly harder. The specificity of these datasets to

their unique locations and conditions makes the standardization that is often required for local and national regulations a challenge. The current state of AI and ML surrounding the careful selection of datasets that are near-perfectly tailored to their unique characteristics is a significant hurdle to overcome in order to make way for local and national regulations to accept these powerful tools on a regulatory and standard basis.

*7.4. Result Comparison Challenges*

This challenge is similar to the issues posed in the Standardization section above but relates specifically to the way researchers conduct and publish their results and measures of success. As was common in almost every paper reviewed in this journal, researchers included some statistical methods to evaluate the accuracy and/or precision of their models. These include but are not limited to the coefficient of correlation (R), the coefficient of determination ($R^2$), the mean average error (MAE), the mean square error (MSE), the root mean square error (RMSE), normalized root mean square error (NRMSE), Pearson correlation coefficient (PCC), Nash–Sutcliffe efficiency coefficient (NSE), and relative error (RE). The challenge is introduced in the fact that there is no standard statistical method employed by researchers using AI/ML models and methods. Coupled with data and reproducibility challenges that lead to incomplete/unpublished raw data (making independent statistical analysis impossible), this makes it difficult to truly compare the results from studies published by researchers using different statistical methods, even when they rely on the same ML models.

*7.5. Explainability Challenges*

The final, but potentially most important challenges regarding the implementation of AI and ML in water applications are the social and legal challenges. We cannot expect these applications to be infallible, just as we cannot expect this from operators and other related experts. It is not a question of if, but a question of when these applications can fail if implemented in critical areas such as drinking-water treatment. The complexity and apparent randomness of some of these models will be difficult to explain to the public at large, but it is increasingly important that the public be equipped to understand the extent and limitations of these systems as much as possible. Legally, it will be difficult to assign blame and enact restitution with the current laws enacted in many localities [130].

## 8. Recommendations for AI/ML Implementation in Water Treatment and Monitoring

It is because of these challenges that future recommendations are necessary and important in ensuring that these powerful models and systems can be implemented to aid some of our most important processes in society. Perhaps the most important recommendations for the future relate to the management of data, the ability to reproduce models in new areas and industries using existing and proven models, the creation/utilization of explainable AI methods and ML models, transparency, and introducing causality into AI and ML constructions.

*8.1. Recommendations for Data Management and Reproducibility*

While many treatment plants and natural systems already have sensing systems to comply with regulatory bodies, regulatory compliance tends to require extended averages. While it is likely that data are originally recorded using small time scales, it is not certain that every area of the water industry has maintained the detailed data that would be necessary to achieve these intelligent systems [131]. Storing data in its most detailed form so that it can eventually be called back and used in training/teaching these systems whenever possible is one way to alleviate this issue. Even systems that are not currently using AI or ML in their daily functions could benefit from creating historical archives of necessary data should the day come that AI or ML would prove useful.

Methods have been proposed to alleviate this data-management issue that has classically prevented many industries from implementing these systems, and also increased the

reproducibility for new areas using existing and proven methods and models. The process known as transfer learning (also referred to as knowledge transfer) is used to translate a pre-trained model from one application to different applications relying only on a relatively small amount of new data.

Transfer learning is ultimately based on the human experience of applying previous knowledge and experience to new problems for faster and better results. Rather than crafting ML models that use traditional learning methods to learn each task from the ground up, transfer-learning techniques seek to use knowledge from past tasks for new tasks, specifically when new tasks lack robust high-quality training data [132]. Methods of transfer learning can be employed when labeled data are available in a target domain (known as inductive transfer learning), when label data are available in a source domain (transductive transfer learning), and when no labeled data exist in both the source and target domains (unsupervised transfer learning) [133].

### 8.2. Recommendations for Transparency and Explainable AI/ML Models and Methods

Transparency is increasingly necessary when utilizing AI and ML, not only so researchers can verify their reproducibility but so that everyone can benefit from the knowledge behind their construction. Humans are generally hesitant to adopt models and methods that are not easily interpretable and trustworthy [134], and with the rise in deep learning and black-box methods (as seen in the models reviewed in this very paper), transparency and explainability are increasingly difficult to come by in AI and ML applications. While divulging the precise methods and construct of deep learning and black-box models between researchers is certainly a viable way of increasing transparency within the academic community, this ultimately still falls short of reaching a larger, more general audience [135]. The concept of explainable AI and ML seeks to do this very thing.

Explainable AI and ML is based on three levels of transparency, namely simulatability, decomposability, and algorithmic transparency [136]. Simulatability, in this case, refers to a human's ability to think (or simulate) a model, and is mainly driven by the model's complexity. For example, models relying on complex rule-based functions or deep, intricate structures, such as ANFIS models, have low simulatability, while models relying on simpler and more explainable rules, such as decision trees and random forests, have high simulatability.

Decomposability refers to the ability to appreciably interpret the individual parts of a model, such as the inputs, parameters, and correlations. It relies on the decomposed model being interpretable by a human without outside help or tools. Algorithmic transparency refers to a person's ability to follow the steps an ML model uses to take an input and create an output. For these reasons, models with deep structures or architecture would have low algorithmic transparency. For a more in-depth discussion of explainable AI and ML, along with methods to make the ML models discussed in this paper more explainable, it is highly recommended that the referenced source from Arrieta et al. be viewed.

### 8.3. Recommendations for Introducing Causality into AI/ML Models and Methods

The ability to overcome poor performance in AI and ML applications due to disturbances between training and testing data can be achieved by introducing causality (i.e., defining the linkage between cause and effect, input and output) between model variables. Stable learning methods, or the ability for a learning algorithm to maintain little to no change when the training data itself is changed, have been used to introduce causality, and come in the form of a variety of different models, namely domain adaptation, domain generalization, and variable decorrelation [137].

Domain adaptation refers to the alteration of a source domain to make it more like that of the target domain [132]. In the case of domain adaptation, labels should be readily available in the source dataset but are usually not readily or ubiquitously available in the target dataset. This method works through simultaneous training using both the source-dataset and target-dataset domains and relying heavily on the labels supplied by

the source dataset to train the unlabeled target dataset. The two datasets should be closely related when using domain adaptation, and data should be readily available from the target dataset.

Domain generalization is similar to domain adaptation, except that it can function with unseen (i.e., assumed unavailable) datasets [138]. This is achieved by using data from related but distinct sources to form a more general model. A commonly used example for a functional domain-generalized model is one that was trained using handmade work such as sketches, cartoons, or paintings but can accurately classify real-life images such as photographs [139].

Variable decorrelation seeks to alleviate the issue of correlation bias, or the perceived illusion that correlation exists between two variables when in fact it is tenuous at best. The process seeks to identify the true significance of a variable. This can be achieved by reweighting samples, thereby removing linkages between variables and specific training data [140]. In some cases, this can be difficult to apply and can reduce sample sizes, but other proposed subsets of variable decorrelation such as decomposed variable decorrelation have been recommended [137].

## 9. Conclusions

Artificial intelligence and machine learning, along with smart technology, can be used to simplify and understand some of the most complex issues facing water-based industries. This review has provided a cross-section and analysis of common ML models, with some AI techniques and smart technologies that have been employed in water-based applications.

ML models and AI methods have adequately optimized, predicted, modeled, and automated some of the most critical applications in water-based industries/operations, including that of water- and wastewater-treatment plants, natural systems, and water-based agriculture. Though many of the studies have been published and reviewed with success, they are not without their own sets of challenges and limitations. Data management, public/legal opinions, reproducibility, and transparency in research are all important hills that must be climbed to further these intelligent applications. While these challenges and limitations are certainly apparent, they do not discount the current research and progress which suggests that ML models, AI methods and smart technologies have important implications and futures for one of our planet's most important resources.

**Author Contributions:** Conceptualization, M.L and X.M; Methodology, M.L and X.M; Formal Analysis, M.L.; Investigation, M.L, X.M, and R.Q; Resources, X.M.; Writing—Original Draft Preparation, M.L.; Writing—Review and Editing, M.L, X.M, and R.Q; Visualization, M.L. Supervision, X.M and R.Q. All authors have read and agreed to the published version of the manuscript.

**Funding:** This research was supported by a grant to the Center for Clean Water Technology (CCWT) from the New York State Department of Environmental Conservation [NYS-DEC01-C00366GG-3350000].

**Institutional Review Board Statement:** Not applicable.

**Informed Consent Statement:** Not applicable.

**Data Availability Statement:** Not applicable.

**Conflicts of Interest:** The authors declare no conflict of interest.

## Abbreviations

AI, Artificial Intelligence; ML, Machine Learning; ANN, Artificial Neural Network; RNN, Recurrent Neural Network; RF, Random Forest; SVM, Support Vector Machine; SVR, Support Vector Regression; FIS, Fuzzy Inference System; ANFIS, Adaptive-Neuro Fuzzy Inference System; GA, Genetic Algorithm; GEP, Genetic Programming; RBF, Radial Basis Function; LSTM, Long Short-Term Memory; CNN, Convolutional Neural Network; k-NN, k-Nearest Neighbor; ELM, Extreme Learning Machine; HW, Hammerstein-Wiener; IoT, Internet of Things; R, Coefficient of Correlation; R2, Coefficient of Determination; MAE, Mean Average Error; MSE, Mean Square Error; RMSE, Root Mean Square Error;

NRMSE, Normalized Root Mean Square Error; RE, Relative Error; MAPE, Mean Absolute Percentage Error; NSE, Nash-Sutcliffe Efficiency Coefficient; ACC, ACCuracy; PCC, Pearson Correlation Coefficient; THM, Trihalomethane; TTHMs, Total Trihalomethanes; HAA, Haloacetic Acid; THAA, Total Haloacetic Acid; DBP, Disinfection By-Product; TOC, Total Organic Carbon; DOC, Dissolve Organic Carbon; DCAA, Dichloroacetic Acid; TCAA, Trichloroacetic Acid; BCAA, Bromochloroacetic Acid; TCM, Trichloromethane; BDCM, Bromodichloromethane; DBCM, Dibromochloromethane; BET, Brunauer, Emmett and Teller; BOD, Biological Oxygen Demand; COD, Chemical Oxygen Demand; DO, Dissolved Oxygen; BGA, Blue-Green Algae; WQI, Water Quality Index; FPR, False Positive Rate.

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
