# Peer review of "A Review on Machine Learning, Artificial Intelligence, and Smart Technology in Water Treatment and Monitoring"

_water, doi:10.3390/w14091384_

Round 1

Reviewer 1 Report

The manuscript is valuable and relevant for water researchers, but some minor revisions are suggested before its publication. Specific comments are listed in the attached pdf file 

Reviewer 2 Report

Dear Authors

This is a very topical and relevant paper. It does highlights how AI/ML can be utilised in Water treatment and monitoring.

However, I have following comments:

1) Abstract - 200 words maximum. I strongly encourage authors to use the following style of abstracts:

Background: Place the question addressed in a broad context and highlight the purpose of the study;

Methods: briefly describe the methodology applied;

Results: summarize the article’s main findings;

Conclusions: indicate the main conclusions or interpretations

2) Introduction Section - It should be a reflection of the Abstract in a broad context. It should define the purpose of work and its significance. Current research needs to carefully reviewed and key publication cited. Briefly mention the main aim of the review and highlight the principal conclusion. Cite some of the relevant papers from the Water journal itself.

3)  Section 3.2 - Please reference following articles 'An Internet of Things Approach for Water Efficiency: A Case Study of the Beverage Factory' 'Improving Water Efficiency in the Beverage Industry with the Internet of Things'

4) Pleas remove Data Availability Statement and Acknowledgement part

5) References - Check the font size. It should be as per MDPI guidelines

Overall, good work.

Regards 
